# High Sensitivity and Wide Range Biomimetic Tactile-Pressure Sensor Based on 2D Graphene Film and 3D Graphene Foam

**DOI:** 10.3390/mi13071150

**Published:** 2022-07-21

**Authors:** Baolin Sha, Xiaozhou Lü, La Jiang

**Affiliations:** 1The 41st Institute of the Fourth Academy of CASC, Xi’an 710025, China; 13709257563@126.com; 2School of Aerospace Science and Technology, Xidian University, Xi’an 710071, China; j15603746663@163.com

**Keywords:** 2D graphene film, 3D graphene foam, tactile sensor, pressure sensor

## Abstract

Bionic electronic skin is a system that simulates human skin and has multiple perceptions. For pressure sensors, high measurement accuracy and wide measurement range restrict each other, and it is difficult to achieve high measurement accuracy and wide measurement range simultaneously. Therefore, the research and application of bionic tactile-pressure sensors are limited due to the mutual constraints of measurement accuracy and range. In this work, a flexible graphene piezoresistive tactile sensor based on a biomimetic structure that utilizes the piezoresistive properties of graphene was reported. The novel tactile-pressure sensor consists of a 2D graphene film tactile sensor and a 3D graphene foam pressure sensor that could achieve high accuracy and a wide-range measurement simultaneously. The testing results show that the measurement range of this sensor was in two intervals of 0–2 N and 2–40 N. For the 0–2 N measurement range, the sensitivity was 472.2 Ω/kPa, the force resolution was 0.01 N, and the response time was less than 40 ms. For the 2–40 N measurement range, the sensitivity was 5.05 kΩ/kPa, the force resolution was 1 N, and the response time was less than 20 ms. The new sensor can realize high-precision and large-scale force measurements and shows great application value in the field of medical instruments and artificial limbs.

## 1. Introduction

With the development of bionic robots and medical devices, it is significant to simulate human perception measurement. As one of the five human senses (vision, touch, hearing, smell, and taste), touch is an important parameter for perceiving the outside world [1]. The pressure sensor can realize the measurement of human touch [2,3,4]. Its working principle can be divided into four categories: capacitive [5,6,7], piezoresistive [8,9,10], piezoelectric [11,12,13,14], and triboelectronic [15,16,17,18,19,20]. Capacitive sensors usually have high sensitivity. Cui et al. fabricated a flexible capacitive tactile sensor based on silver electrodes and Polydimethylsiloxane (PDMS), with a sensitivity of up to 19.8 kPa^−1^ [21]. Wan et al. proposed an ultrasensitive pressure sensor with graphene as an electrode. The response time of the sensor can reach 100 ms, and the sensitivity of the sensor can reach 0.8 kPa^−1^ [22]. Ntagios et al. developed a soft tactile sensor by using the 3D print method, and the soft tactile sensor exhibited a stable response with a sensitivity of 0.00348 kPa^−1^ for pressure <10 kPa and 0.00134 kPa^−1^ for higher pressure [23]. Yang et al. demonstrated a 3D microconformal graphene electrode for a flexible capacitive pressure sensor and the developed capacitive pressure sensor with a high sensitivity of 3.19 kPa^−1^, a fast response time of 30 ms, and an ultralow detection limit of 1 mg [24]. Kang et al. proposed a highly sensitive capacitive pressure sensor based on a porous structure of PDMS, and the developed sensor resulted in high-performance pressure sensors with high sensitivity of 0.63 kPa^−1^ and extremely low-pressure detection of 2.42 Pa [25]. From the above cases, the capacitive sensor has a high sensitivity but a small measurement range. At the same time, the capacitive sensor has some shortcomings, such as capacitive sensors have large impedances, which lead to them being sensitive to the parasitic capacitance of coaxial cables, and the measurement error is large [26]. Piezoelectric self-sensing is a promising technique that has captured a broad range of attention since its origin. The piezoelectric sensor has good dynamic performance, but its static performance is poor and is prone to drift [27,28]. Spanu et al. proposed a high-sensitivity tactile sensor based on a floating gate organic transistor, an organic charge modulation polyethylene terephthalate (FET) coupled with polyvinylidene difluoride (PVDF), and the lowest pressure can be measured at 300 Pa [29]. The piezoresistive sensor has a simple structure, stable working state, and relatively low cost, so it is widely used. Rinaldi et al. studied PDMS foam sensors coated with multilayer graphene nanosheets, and pressure of about 70 kPa can be measured, corresponding to a sensitivity of 0.23 kPa^−1^ [30].

Traditional tactile sensors are all made of a single pressure sensor. For example, the pressure sensor proposed by Wan et al. has a sensitivity of up to 0.8 kPa^−1^, high accuracy, and a measurement range of up to 1.2 kPa [22]. Huang et al. incorporated highly conductive polyaniline (PANI) polymers, graphene nanoplates (GNPs), and a small amount of silicone rubber (SR) onto elastic Lycra fabrics by spin coating to fabricate strain sensors [31]. The strain range is up to 40%, but its accuracy is relatively low [31]. It can be seen from the above analysis that traditional sensors are contradictory in terms of measurement range and accuracy, and it is difficult to maintain a large measurement range while satisfying high precision at the same time. Moreover, the haptic function of the human body can not only measure the high-precision micro pressure but also realize the perception of a large measurement range. Human tactile sensation differs by 2 to 3 orders of magnitude in terms of measurement range and accuracy, so it is difficult for a single sensor to meet the above requirements.

Graphene is a star material as a two-dimensional “aromatic” monolayer of carbon atoms with sp^2^ atomic configuration that exhibits exceptionally physical properties [32,33] and is widely used in the sensing area [34,35,36]. In this work, to overcome the contradiction between the measurement range and accuracy of the pressure sensor, this project proposes a graphene-based multi-sensitive cell tactile sensor. The sensor utilizes the piezoresistive properties of graphene. It combines two sensors based on the bionic structure of the human body, which can meet the requirements of simultaneous measurement with high precision and a wide measurement range. This sensor used 2D graphene/PET film as a small-range high-precision tactile module and graphene/polyurethane (PU) sponge foam as a wide-range pressure module. The results show that the measurement range of this sensor was in two intervals of 0–2 N and 2–40 N. For the 0–2 N measurement range, the sensitivity was 472.2 Ω/kPa, the force resolution was 0.01 N, and the response time was less than 40 ms. For the 2–40 N measurement range, the sensitivity was 5.05 kΩ/kPa, the force resolution was 1 N, and the response time was less than 20 ms. The new sensor can realize high-precision and large-scale force measurements and shows great application value in the field of medical instruments and artificial limbs.

## 2. Methods and Materials

### 2.1. Structure Design and Working Principle of the Sensor

The sensor proposed in this paper was a biomimetic flexible tactile sensor inspired by human skin, so the sensor should conform to human bionic parameters as much as possible under the premise of ensuring flexibility. The previous research shows that the tactile perceived by the skin of the human hand should have a force measurement sensitivity of 0.01 N, a force measurement range of 0~10 kPa, and a geometric resolution of about 2 mm [37]. Pressure perceived by the skin of the human hand should have a sensitivity of 1 N, a force measurement range of 0~220 kPa, and a 1 cm geometric resolution. Figure 1a was the schematic diagram of the sensor structure. The sensor consists of upper and lower sensor units. The upper sensing units were a layer of 2D graphene film tactile sensors, and the lower sensing units were a 3D graphene foam pressure sensor. The tactile sensor consists of four 2D graphene/PET films with a size of 3 mm × 3 mm and Flexible Printed Circuit Board (FPCB) electrodes to form four tactile sensing units. Each tactile sensing unit corresponds to a group of electrodes as output, the unit spacing is 3 mm, and the spatial resolution of the tactile sensing unit is 3 mm. The pressure sensor consisted of a top electrode, a PDMS support structure, a 3D graphene foam, and a bottom electrode. The field-shaped PDMS support structure has 4 square hollow grooves, and each hollow groove is embedded with graphene foam with a size of 3 mm × 3 mm × 5 mm.

The measurement circuit schematic of the sensor is shown in Figure 1b,c, which consists of two parts. Since the four graphene film sensors of the upper tactile sensor were connected to independent electrode outputs, they can be regarded as four independent variable resistors, as shown in Figure 1b. The lower layer is a pressure sensor, due to the “sandwich” structure. The upper and lower electrodes are connected to 4 electrodes separated by PDMS, as shown in Figure 1c. The working principles of tactile and pressure sensors both use the piezoresistive effect of graphene. That is, the C-C bond of the graphene structure is broken when subjected to pressure, resulting in a change in the resistivity of the 2D graphene film and 3D graphene foam, which in turn causes graphene changes in resistance of films and foams.

### 2.2. Fabrication of the Tactile Sensor

Figure 2 shows the fabrication processes of the sensitive layer of the tactile sensor. Firstly, copper foil (Cu) with a thickness of 50 μm was selected as the substrate, and graphene was grown on its surface by the chemical vapor deposition (CVD) method. Thus, the Cu with graphene thin film(Cu/Graphene) was obtained, as shown in Figure 2a-I. Secondly, the polymethyl methacrylate (PMMA) solution was spin-coated on the surface of the Cu/Graphene, and the sandwich structure of Cu/Graphene/PMMA was obtained, as shown in Figure 2a-II. Here, the PMMA supported the two-dimensional structure of graphene film, which facilitates the subsequent copper foil etching and graphene transfer operations. Thirdly, the copper of the sandwich structure of Cu/Graphene/PMMA was etched using HCL and CuSO_4_ solution, and the Graphene/PMMA was obtained, as shown in Figure 2a-III. Finally, PET with a thickness of 100 μm was selected as the substrate, graphene film was transferred to the surface of PET using the wetting transfer method, and PMMA film was removed by cleaning with acetone; thus, Graphene/PET film was obtained, as shown in Figure 2a-IV. Figure 2b shows the physical preparation processes of the graphene film. The plane view image of the prepared graphene film is shown in Figure 2c.

Figure 3 shows the fabrication processes of the tactile sensor. Firstly, the obtained graphene/PET film was cut into 3 mm × 5 mm, as shown in Figure 3a(I–II). Secondly, the cut graphene/PET film of 3 mm × 5 mm was attached to the electrode surface using silver glue, and the graphene was in contact with the electrodes. The left and right sides of each graphene film were attached to electrodes with a width of 1 mm; therefore, the final size of a single tactile cell unit was 3 mm × 3 mm, as shown in Figure 3a(III–IV). Finally, when the conductive glue is well cured in a vacuum drying oven at a temperature of 80 °C, a layer of PDMS film with a thickness of 200 μm was attached to the top surface of the entire tactile sensor array for protecting the sensitive cells. Figure 3b is the picture of the fabricated tactile sensor. It can be seen that the sensor is small in size and has strong overall flexibility. Since the PDMS film was used as the packing material, the graphene film was robust and not easy to fall off when pressure was applied to the surface of the device.

### 2.3. Fabrication Processes of the Pressure Sensor

Figure 4 shows the fabrication processes of the pressure sensor. Firstly, the PU foam with a hardness of 27 was cut into a size of 3 mm × 3 mm × 5 mm and washed with absolute ethanol and deionized water sequently, then put into a vacuum drying box to dry for 15 min. Secondly, the graphene oxide (GO) suspension was prepared using graphene oxide and deionized water with a ratio of 2 mol/mL. The clean PU foam was immersed into the GO suspension, and the GO solution was adequately absorbed and then dried at a low temperature of 60 °C for 45 min. Thirdly, the Pu foam that fully absorbed GO (GO/PU) was immersed in the vitamin C (VC) solution at a high temperature of 80 °C to reduce the GO/PU foam to obtain the rGO/PU foam as shown in Figure 4a. Finally, the obtained rGO/PU foam was washed and dried and compressed several times to obtain a stable internal structure, as shown in Figure 4b. The plane view SEM image of the prepared 3D graphene foam is shown in Figure 4c.

Our previous experiment results show that the modulus of the PDMS with a ratio of 25:1 was close to that of graphene/PU foam and was suitable as a support structure for the foam. The PDMS support structure mold was fabricated using a 3D printer machine, and the design of the mold is shown in Figure 4d. The model was designed with four cuboid columns on the base, and the size of each column was 5 mm, 3 mm, and 3 mm in height, long and wide, respectively. The column was 1 mm away from the boundary, and the distance between each column was 2 mm. The preparation processes of PDMS were as follows. PDMS prepolymer with a ratio of matrix material and curing agent of 25:1 was adequately stirred and degassed in a vacuum dry oven and then injected into the prepared mold. The mold with PDMS mixed solution was put into a vacuum oven and cured at a high temperature of 80 °C for 45 min. Finally, the PDMS was peeled from the mold, and the PDMS support structure was obtained, as shown in Figure 4d. Figure 4e is the picture of the fabricated PDMS structure.

Figure 4f shows the fabrication processes of the pressure sensor. Firstly, the prepared 3 mm × 3 mm × 5 mm graphene/PU foam was inserted into the “+” shaped PDSM support structure with a tweezer and adjusted so that there was no gap between the graphene foam and the inner wall of the support structure. Then, two pieces of copper foil were cut into a size of 10 mm × 12 mm and conductive silver glue was applied to the exposed ends of the graphene foam, and a little adhesive was applied to the surface of the support structure and attached to the copper foil to it. Afterward, the sensor was put into a vacuum oven at a high temperature of 80 °C to cure the conductive glue and adhesive, and then two copper wires were connected to the copper foil; thus, the pressure sensor was successfully fabricated. The actual picture of the pressure sensor is shown in Figure 4g.

### 2.4. Assemble the Tactile-Pressure Sensor

Figure 5 shows the flow chart and physical diagram of the preparation principle of the tactile-pressure sensor. The preparation process was as follows. Firstly, the bottom of the FPCB of the tactile sensor based on the 2D graphene film was coated with PDMS. Then, a little adhesive was coated on the top of the copper foil of the pressure sensor based on 3D graphene foam. Secondly, the two sensors were pasted together and put into a vacuum drying oven to cure the adhesive at a high temperature of 80 °C. While curing the adhesive, a heavy object of 2 N was placed on the top of the sensor so that the two sensors could fit tightly. After the PDSM was cured, a flexible bionic tactile-pressure sensor based on 2D/3D graphene was obtained, and the actual sensor picture is shown in Figure 5c.

### 2.5. Experimental

Figure 6 shows the experimental platform used in this work, a digital pressure gauge and a digital SourceMeter were used to measure the changing of pressure and resistance, respectively. The tactile-pressure sensor unit was placed on the three-dimensional force experiment platform and the digital pressure gauge on the longitudinal pressure arm by applying longitudinal pressure to it. The other end of the tactile-pressure sensor was connected to the digital SourceMeter (Keithley, 2450). The output resistance of the corresponding tactile-pressure sensor was obtained by recording the readings of the SourceMeter in real-time. Figure 6a was the illustration schematic of the testing platform, and Figure 6b was the system’s photo.

## 3. Results and Discussion

Figure 7 shows the output properties of the flexible bionic tactile-pressure sensor proposed in this study. The resistance of the tactile sensor increased with the applied force and the pressure sensor exhibited an opposite changing trend. The sensing range of the tactile and pressure sensor was 0–2 N and 2–40 N, respectively, as shown in Figure 7a,b. The least-squares method was used to fitting the relationship between the input force and the output resistance of the sensors, as shown in Figure 7c,d. The least-squares fitting results of the tactile and pressure sensor, as shown in Equations (1) and (2), were as follows:*R* = 0.00407 F + 1.9337(1)
*R* = −0.45622 F + 26.2219(2)

The sensitivity was the ratio of the output increment of the sensor to the input increment during measurement, which was expressed in Equation (3), as follows:(3)$$S=\frac{\Delta R}{\Delta F}$$

The sensitivity of a linear measurement system can be calculated from the slope of the static characteristic curve. The sensitivity of the tactile and pressure sensor can be calculated by using Equation (3), which was 472.2 Ω/kPa and 5.50 kΩ/kPa, respectively. The demonstrative experiment of the sensor was shown in Appendix A.

Figure 8 shows the repeatability and hysteresis properties of the tactile and pressure sensor, and 1#, 2# and 3# represent the first, second and thired sensors. The input force and output resistance relationship of the sensor was recorded and graphed by repeating the measurements on the tactile and pressure sensor three times, and the data are shown in Figure 8a,b. The repeatability refers to the degree to which the output curve deviates from multiple measurements of the sensor’s full scale and is an important feature for a sensor. The repeatability error δ_R_ was usually calculated with the standard deviation. The calculation results show that the repeatability error of the tactile and pressure sensor proposed in this study were 16.55 and 10.48%, respectively, and the tactile-pressure sensor exhibited a good repeatability performance.

The hysteresis of a sensor refers to the non-overlapping phenomenon of the output characteristic curve when the sensor is in the forward (increase in input) and reverse (increase in input) stroke. The hysteresis curves of the tactile and pressure sensor in this study are shown in Figure 8c,d, respectively. The maximum hysteresis was used to evaluate the hysteresis of a sensor, which is expressed in Equation (4) as follows:(4)$$\delta_{H}=\frac{\Delta H_{\max}}{Y_{FS}}\times 100\%$$

$\delta_{H}$ is the maximum hysteresis reference error, $\Delta H_{\max}$ is the output maximum absolute deviation, and $Y_{FS}$ is the full-scale value of the sensor. The hysteresis of the tactile and pressure sensor proposed in this study was 4.7% and 7.23% by using Equation (4), respectively.

The dynamic response property refers to the response time to changes in the sensor’s input and represents the speed at which the measurement system tracked the input variable parameters. The response time was regarded by many researchers as the most important indicator of dynamic performance and can be used to accurately measure the delay relationship between output and input. Figure 9 shows the dynamic response property of the tactile and pressure sensor in this study. The acquisition frequency of the output resistance was 100 Hz, and it can be obtained from the data in Figure 9a that the response time of the tactile sensor was about 30 ms when the force was applied and about 40 ms when the force was released. The response time of the pressure sensor was less than 20 ms, which can be seen in Figure 9b.

Figure 10 shows the working mechanism of the tactile-pressure sensor. When an external force was applied to the tactile sensor, the C-C bonds were broken, and the two-dimensional structure of graphene was destroyed. Thereby, the conductive patch was reduced, the resistance was increased, and the resistance change was linearly related to the applied external force to a certain extent, as shown in Figure 10a. For the pressure sensor, when no external force was applied, a layer of graphene was attached to the inner wall and surface of PU foam and had a stable structure. When an external force was applied, the graphene patches on the inner wall of the foam were damaged. While continuously applying an external force to the pressure sensor, the collapsed parts of the foam will come into contact, thereby increasing the number of conductive patches, and the resistance was decreased, as shown in Figure 10b.

## 4. Conclusions

In this paper, we present a flexible biomimetic tactile-pressure sensor with high precision and a large measurement range by using 2D graphene film and 3D graphene/PU foam. The tactile-pressure sensor consisted of two layers of structures, and the size was 10 mm × 10 mm × 6 mm in longitude, width, and thickness. The fabrication process, output properties, repeatability, and hysteresis properties were systematically studied. The working mechanism of the tactile-pressure sensor was proposed and discussed. The results exhibit a sensitivity of 472.2 Ω/kPa in the range of 0–2 N, a force resolution of 0.01 N, and a response time of less than 40 ms for the tactile sensor. The sensitivity can reach 5.05 kΩ/kPa, a force resolution of 1 N. The response time was less than 20 ms for the pressure sensor in the measurement range of 2–40 N. The sensor spaned a measurement range of 4 orders of magnitude. These experimental results show that the flexible bionic tactile-pressure sensor designed in this paper can be used to measure both small pressure and large pressure. This has certain reference value for fields such as intelligent robots, electronic skin, and biomedicine.

## Figures and Tables

**Figure 1 micromachines-13-01150-f001:**
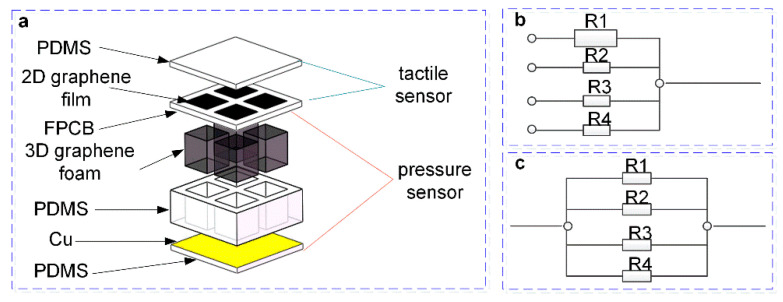
The structure of the designed sensor and measurement circuit schematic. (**a**) The structure of the designed sensor. (**b**) The measurement circuit schematic of the tactile sensor. (**c**) The measurement circuit schematic of the pressure sensor.

**Figure 2 micromachines-13-01150-f002:**
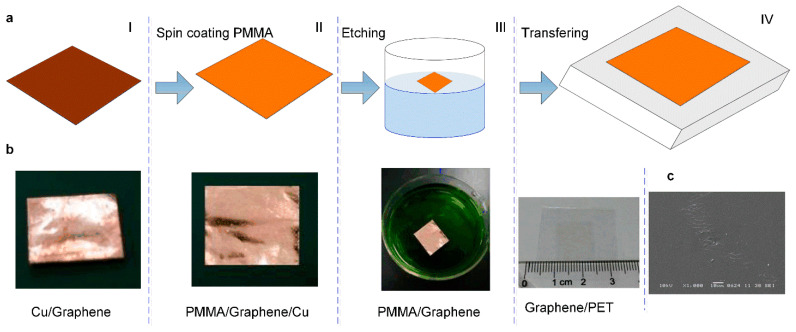
The preparation processes of graphene film. (**a**) The schematic of the fabrication processes of graphene film. (**b**) The physical preparation processes of the graphene film. (**c**) The plane view SEM image of the prepared graphene film.

**Figure 3 micromachines-13-01150-f003:**
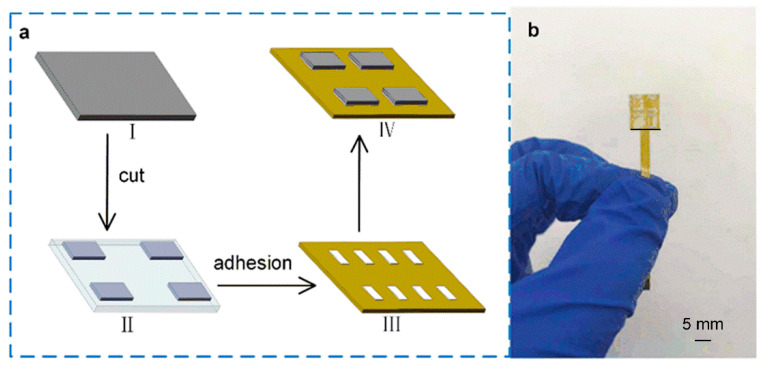
The fabrication processes of the tactile sensor and the photo of the tactile sensor. (**a**) The fabrication processes of the tactile sensor. (**b**) The photo of the fabricated tactile sensor.

**Figure 4 micromachines-13-01150-f004:**
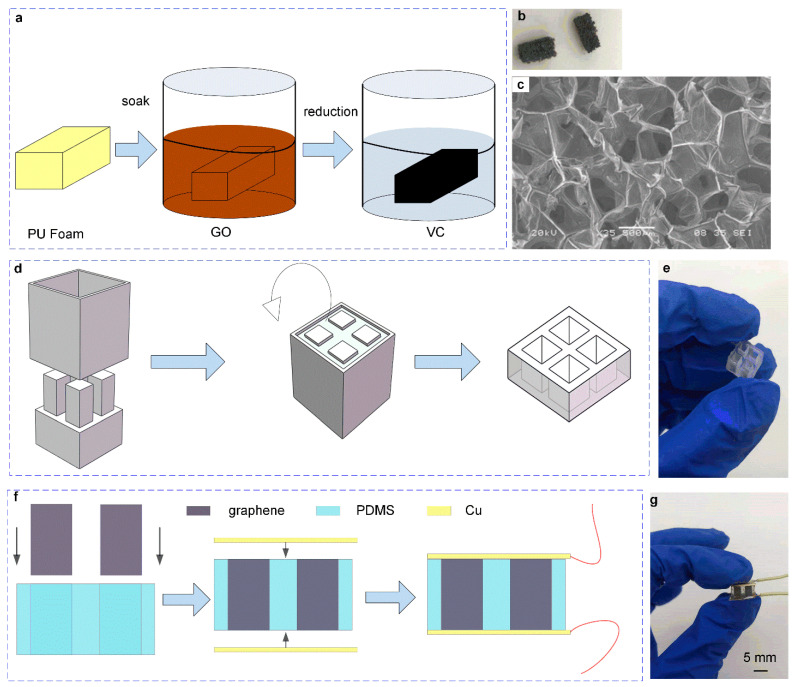
The fabrication processes of the pressure sensor. (**a**) The fabrication processes of the graphene foam. (**b**) The photo of fabricated graphene foam. (**c**) The plane view SEM image of prepared 3D graphene foam. (**d**) The fabrication processes of the support structure of the pressure sensor. (**e**) The photo of the fabricated support structure. (**f**) The fabricated processes of the pressure sensor. (**g**) The photo of the fabricated pressure sensor.

**Figure 5 micromachines-13-01150-f005:**
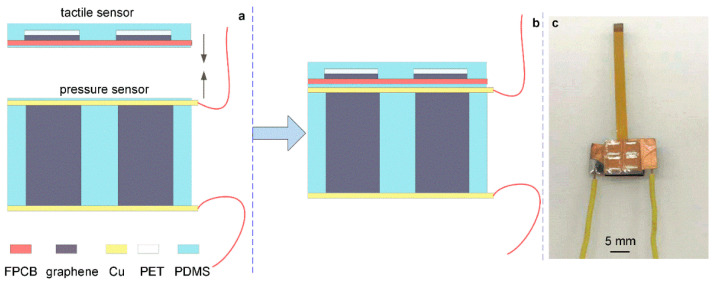
The flow chart and physical diagram of the assembling processes of the tactile-pressure sensor. (**a**) The assemble processes of the tactile-pressure sensor. (**b**) The schematic of the assembled tactile-pressure sensor. (**c**) The photo of the fabricated tactile-pressure sensor.

**Figure 6 micromachines-13-01150-f006:**
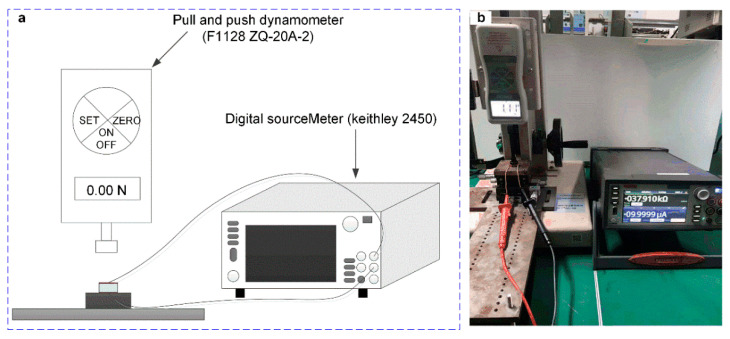
The testing platform. (**a**) The schematic diagram of the force-resistance testing platform. (**b**) The photo of the testing platform.

**Figure 7 micromachines-13-01150-f007:**
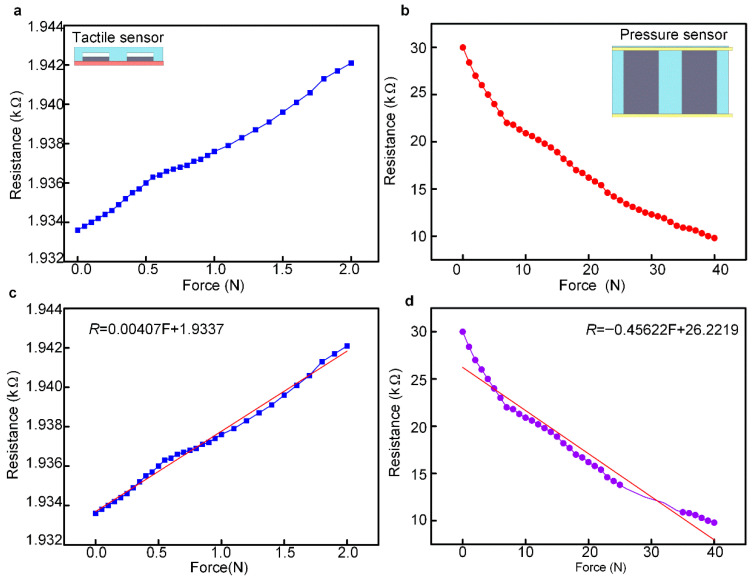
The output properties of the flexible bionic tactile-pressure sensor. (**a**) The relationship between the force and output resistance of the tactile sensor. (**b**) The relationship between the force and output resistance of the pressure sensor. (**c**) The fitting result of the force and output resistance of the tactile sensor by using the least-squares method. (**d**) The fitting result of the force and output resistance of the pressure sensor by using the least-squares method.

**Figure 8 micromachines-13-01150-f008:**
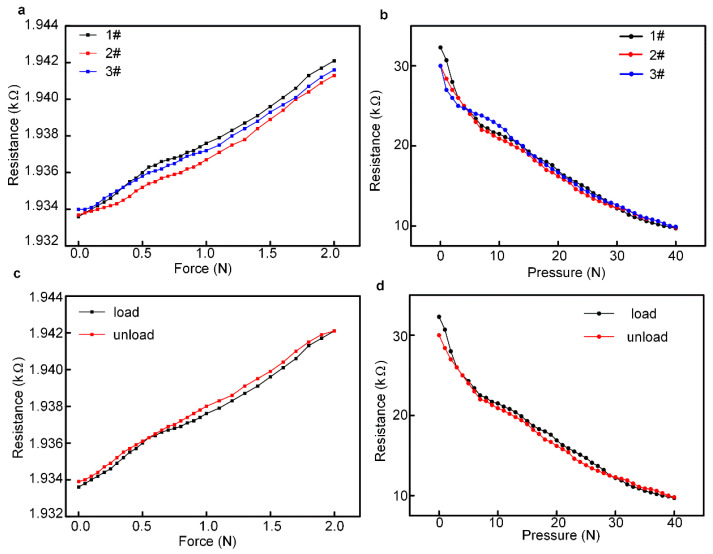
The repeatability and hysteresis properties of the tactile and pressure sensor, and 1#, 2# and 3# represent the first, second and thired sensors. means the (**a**) The repeatability property of the tactile sensor. (**b**) The repeatability property of the pressure sensor. (**c**) The hysteresis property of the tactile sensor. (**d**) The hysteresis property of the pressure sensor.

**Figure 9 micromachines-13-01150-f009:**
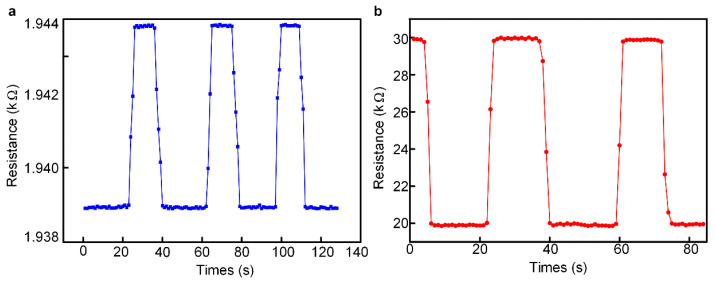
The dynamic response property of the tactile and pressure sensor. (**a**) The response time of the tactile sensor. (**b**) The response time of the pressure sensor.

**Figure 10 micromachines-13-01150-f010:**
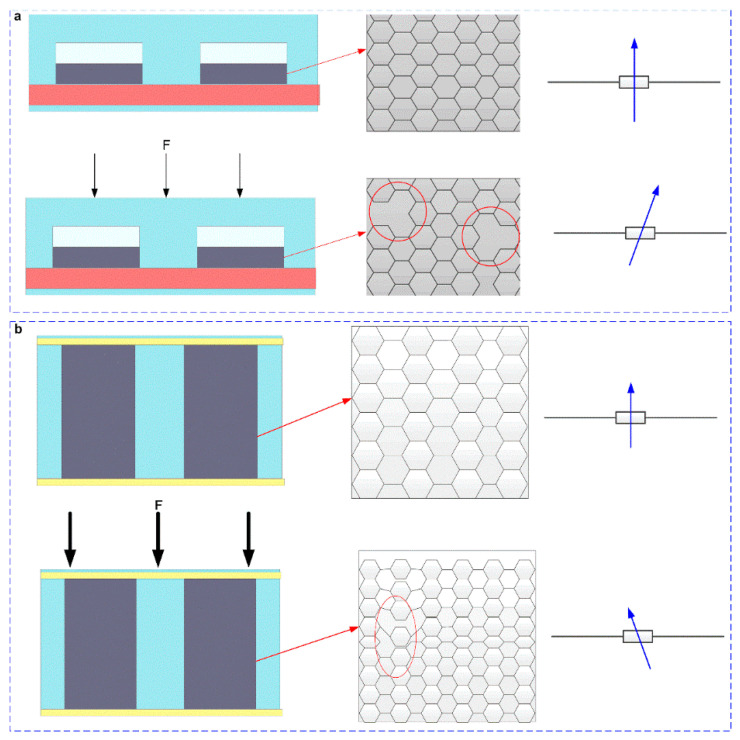
The working mechanism of the tactile-pressure sensor. (**a**) The working mechanism of the tactile sensor. (**b**) The working mechanism of the pressure sensor.

## Data Availability

Not applicable.

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
