# Peer review of "High Sensitivity and Wide Range Biomimetic Tactile-Pressure Sensor Based on 2D Graphene Film and 3D Graphene Foam"

_micromachines, 2022, doi:10.3390/mi13071150_

Round 1

Reviewer 1 Report

The Manuscript by Sha et al. reports a flexible graphene piezoresistive tactile sensor based on biomimetic structure that utilizes the piezoresistive properties of graphene. The as prepared sensors show a very short response time. The idea is interesting and the paper is well-written. In my opinion, the paper could be published after a minor revision. 

1. The micromorphology of the as-prepared materials should be measured.

2. The properties of the sensors are required to compare with the reported works.

3.  More references are required.

4. The demonstrative experiments are needed.

Reviewer 2 Report

In this work, the authors proposed a high sensitivity and wide range pressure sensor based on 2D graphene film and 3D graphene foam. The developed sensor exhibited good performance and this work is valuable. Thus, I recommend the manuscript be accepted and promptly published. The following questions need to be addressed:

1. The authors need to check the express style of the physical parameter and a blank between the number and units.

2. The authors are suggested to give more detailed writing for the discussion.

3. The author should check the reference carefully and some mistakes in the format of  the reference need to be revised:

e.g. reference 4, the paper number, or the page should be given.

4. The authors are suggested to polish the English writing.

Reviewer 3 Report

The authors introduce a combined high sensitivity-short range graphene film and low sensitivity- long range graphene foam pressure sensor for tactile application. They provide theoretical working principles, fabrication process and the experimental results for this sensor. The concept of combining these two sensors in the same module seems to be novel and their experimental results support the their thesis in designing a high sensitivity- long range pressure sensor for tactile application. However this manuscript (MS) needs extensive editorial and language edits before being acceptable for publication. Please address the following issues in your revised MS,

- The abstract as well as introduction is unreadable and needs to be edited/re-written extensively
- All the figures' captions need to include explanation on what each part of the figure shows. Figure 7 is a good example on what information the caption should convey.
- The introduction part needs to be bolstered with more references to prove your thesis that high sensitive/long range pressure sensors are a challenging research area.
- You need to use the long format of the acornym you use for the first time (e.g. FPCB, PU, PET, PMMA and etc)
- L33: You can make this judgment "the capacitive sensor has a high sensitivity..." by the (incomplete) information you have collected from only 2 references. Please expand this section. Provide all the tactile capacitive pressure sensors in the literature and their sensitivity and range.
- L35 please provide a reference on the capacitive sensor shortcomings
- L36 stray and parasitic capacitance are the same in literature use only one
- L36 please provide a reference for your claim "The piezoelectric sensor has good dy- 35 namic performance, but its static performance is poor and is prone to drift"
- L44 the dimension of the sensitivity is incorrect. 0.8% kPa-1?
- L48 please provide references and evidence with actual numbers (accuracy) for your claim "...but its accuracy is relatively low"
- L76 Please provide reference for previous research works you have provided the tactile force/pressure perceived by human skin.
- In figure 1 please label each component (layer) of the sensor
- Figure 2's caption is incorrect. Please update
- Please add a scale bar to all you figures in which you're showing the sensor or part of the sensor
- Please provide information on the thickness of all the layers
- I suggest combining figures 2 and 3 and adding a cartoon of the side profile for each fabrication step (something similar to figure 4-f)
- L 111 how was the graphene film transferred to PET? and how was the PMMA removed?
- L 121 what's the curing process?
